# Detecting stoichiometry of macromolecular complexes in live cells using FRET

Manu Ben-Johny[1], Daniel N. Yue[1] & David T. Yue[1]

The stoichiometry of macromolecular interactions is fundamental to cellular signalling yet challenging to detect from living cells. Fluorescence resonance energy transfer (FRET) is a powerful phenomenon for characterizing close-range interactions whereby a donor fluorophore transfers energy to a closely juxtaposed acceptor. Recognizing that FRET measured from the acceptor's perspective reports a related but distinct quantity versus the donor, we utilize the ratiometric comparison of the two to obtain the stoichiometry of a complex. Applying this principle to the long-standing controversy of calmodulin binding to ion channels, we find a surprising $Ca^{2+}$-induced switch in calmodulin stoichiometry with $Ca^{2+}$ channels—one calmodulin binds at basal cytosolic $Ca^{2+}$ levels while two calmodulins interact following $Ca^{2+}$ elevation. This feature is curiously absent for the related Na channels, also potently regulated by calmodulin. Overall, our assay adds to a burgeoning toolkit to pursue quantitative biochemistry of dynamic signalling complexes in living cells.

[1] Calcium Signals Laboratory, Department of Biomedical Engineering, The Johns Hopkins University School of Medicine, Ross Building, Room 713, 720 Rutland Avenue, Baltimore, Maryland 21205, USA. Correspondence and requests for materials should be addressed to M.B.-J. (email: manu@jhmi.edu).

The dynamic association of biological macromolecules constitutes a fundamental mode of cellular signalling. In this regard, stoichiometry represents a critical parameter essential for the elucidation of mechanisms underlying such molecular interactions, for evaluation of their biological pertinence and for defining their pathological roles. Traditionally, in vitro biochemical methods such as analytical centrifugation[1], equilibrium sedimentation[2], isothermal calorimetry[3] and mass spectrometry[4] have been applied to deduce the stoichiometric relations of components within purified protein complexes. However, large macromolecular signalling complexes such as those of voltage-gated ion channels are often not amenable for such in vitro reconstitution[5] and establishing the stoichiometry of channel interacting signalling molecules has been notoriously challenging though long desired[6]. Therefore, a general live-cell method to define stoichiometric relations for such signalling complexes would facilitate the study of macromolecular quaternary organization and elucidate mechanisms underlying normal and pathological molecular functions.

One prominent example of uncertainty concerns the binding of the ubiquitous $Ca^{2+}$-binding protein calmodulin (CaM) to the voltage-gated $Ca^{2+}$ ($Ca_V$) and Na ($Na_V$) channel superfamily[7,8]. For $Ca_V$ channels, CaM serves as a constitutive component[9] eliciting multiple functional roles including feedback regulation of channel gating[10–12], modulation of cell surface trafficking[13,14] and $Ca^{2+}$-dependent signalling to various local enzymes[15,16]. Even so, the stoichiometry of CaM in the $Ca_V$ channel complex has remained controversial for over a decade[7]. Functional studies have argued that a single CaM is both necessary and sufficient to modulate channel gating[17]. By contrast, structural and biochemical studies have identified CaM binding to several short peptides derived from channel cytosolic domains, suggesting that multiple CaM may yet interact with the holochannel complex[18–20]. Similar controversies have clouded the understanding of CaM regulation of related $Na_V$ channels[8,21,22]. Accordingly, a robust method to determine stoichiometry of molecular signalling complexes in live cells as they perform their cellular functions would greatly aid the resolution of such controversies and may further reveal novel insights for diverse biological systems.

To this end, fluorescence resonance energy transfer (FRET) is a powerful spectroscopic phenomenon to interrogate close-range molecular interactions and to track their dynamics[23]. Typically, upon photoexcitation a fluorophore may de-excite through direct emission of a photon. However, in the presence of an appropriate closely juxtaposed acceptor fluorophore, the donor may de-excite through energy transfer to the acceptor, understood as long-range dipole-dipole coupling (Fig. 1a). The excited acceptor may then release a photon, though possibly with a red-shifted spectrum. This non-radiative transfer of energy is termed FRET and the measurement of the propensity for energy transfer, reported as FRET efficiency, depends upon the spectral properties of the two fluorophores and their relative spatial arrangement, including distance and orientation[20,24]. With the advent of genetically encoded fluorescent molecules, this method has found widespread biological applications as a spectroscopic atomic-scale ruler[25], for the development of biosensors[26], and for in situ detection of biomolecular interactions[9,27] (FRET 2-hybrid assay). Here, we exploit a fundamental asymmetry in the FRET measurements to determine the stoichiometry of macromolecular interactions within living cells. We demonstrate the utility of this method by addressing the long-standing controversy of stoichiometry of CaM binding to voltage-gated ion channels. We find that the L-type $Ca_V$ channels interact with a single CaM at resting $Ca^{2+}$ levels; however, upon cytosolic $Ca^{2+}$ elevation, an additional $Ca^{2+}$-bound CaM is recruited to the channel complex.

By contrast, for the related skeletal muscle $Na_V$ channel isoform, our assay reveals a 1:1 stoichiometry of CaM interaction both at basal and at elevated $Ca^{2+}$ conditions. These findings bear novel insights on CaM signalling to both the $Ca_V$ and $Na_V$ channel complexes. Importantly, the FRET-based assay represents a simple and robust method to deduce the stoichiometry of biological complexes within the milieu of live cells.

## Results

**Deducing stoichiometry from FRET efficiencies.** The process of FRET alters two key features of the total fluorescence emission spectrum of the bound donor-acceptor complex: (1) quenching of the fluorescence intensity of the donor and (2) increase in the fluorescence intensity of the acceptor[27,28] (Supplementary Fig. 1). These spectral changes imply that FRET efficiency can be determined using two distinct metrics[27]: (1) a donor-centric measure that reports the fractional reduction in the donor's fluorescence intensity as a result of FRET[29–32] and (2) an acceptor-centric measure that quantifies sensitized emission or the fractional enhancement in the acceptor's fluorescence intensity due to FRET[9,28,30,32–35]. These distinct measures depend on the number of donors and acceptors in the complex and can therefore be exploited to determine the stoichiometry of the underlying binding interaction. In recent years, several experimental strategies[9,29–31,33–35] have been developed to quantify both acceptor and donor-centric metrics of FRET efficiencies in live cells despite challenges posed by significant spectral overlap between popular fluorescent protein pairs such as the enhanced cyan fluorescent protein (ECFP) and the enhanced yellow fluorescent protein (EYFP). Of note, the $3^3$-FRET method determines the acceptor-centric metric of FRET efficiency ($E_A$) by unscrambling sensitized emission from fluorescence measurements through three distinct filter cubes—termed CFP, YFP and FRET cubes[9]. Similarly, the E-FRET method determines donor-centric metric of FRET efficiency ($E_D$) from the same three fluorescence measurements albeit using a different formula that estimates the fractional quenching of the donor molecule[29,32].

For 1:1 stoichiometry of donor-acceptor interaction, the maximal FRET efficiencies estimated by both $3^3$-FRET (acceptor-centric) and E-FRET (donor-centric) methods correspond to the time-independent transition probability of fluorescence energy transfer from the only donor to the only acceptor in the complex and therefore must be equal to each other[9,30] (Fig. 1a,b, Supplementary Fig. 2; Supplementary Note 1). However, if the bound complex has multiple donors or acceptors, then the $3^3$-FRET method reports the expected number of energy transfer events per acceptor in the complex given that all donors are excited (Fig. 1c,d, Supplementary Fig. 3; Supplementary Note 1).

$$E_{A,max} = \frac{1}{n_A} \sum_{i=1}^{n_D} \sum_{j=1}^{n_A} E_{ij} \qquad (1)$$

Here $E_{A,max}$ corresponds to the maximal $3^3$-FRET efficiency assuming all acceptor molecules are bound, $n_A$ is the number of acceptor molecules in the complex, $n_D$ is the number of donor molecules in the complex, and $E_{ij}$ is the time-independent transition probability of energy transfer (or pairwise FRET efficiency) between $i$th donor and $j$th acceptor. By contrast, the E-FRET method estimates the expected number of energy transfer events per donor molecule in the complex given that all such donor molecules are excited[36] (Fig. 1c,d; Supplementary Fig. 3; Supplementary Note 1).

$$E_{D,max} = \frac{1}{n_D} \sum_{i=1}^{n_D} \sum_{j=1}^{n_A} E_{ij} \qquad (2)$$

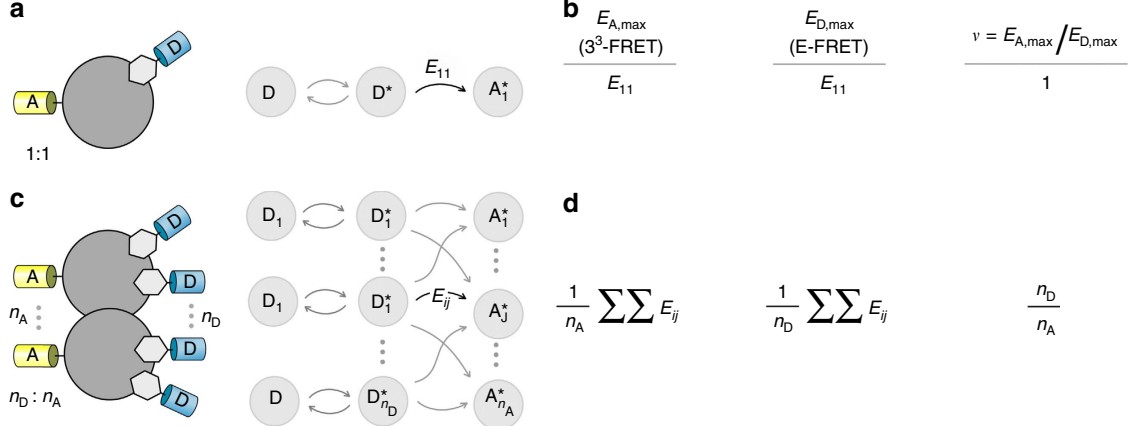

**Figure 1 | Theoretical scheme for deducing stoichiometry from FRET efficiencies.** (**a**) Left, cartoon illustrates 1:1 stoichiometry interaction of two binding partners tagged with donor and acceptor fluorophores. Right, conceptual scheme illustrates FRET between the excited donor ($D^*$) and the bound acceptor ($A^*$). $E_{11}$, the probability of energy transfer. (**b**) Both $3^3$-FRET and E-FRET methods report maximal FRET efficiency as $E_{11}$ yielding stoichiometry ratio $v = 1$. (**c**) Left, cartoon illustrates a multimeric complex containing $n_D$ donor and $n_A$ acceptor molecules. The probability of energy transfer from $i$th donor to $j$th acceptor is denoted $E_{ij}$. (**d**) For multimeric complexes, the maximal $3^3$-FRET ($E_{A,max}$) and E-FRET ($E_{D,max}$) may differ. The stoichiometry ratio $v = E_{A,max}/E_{D,max}$ reports the ratio of number of donors to acceptors in the complex ($n_D/n_A$).

Note that $E_{D,max}$ corresponds to the maximal E-FRET efficiency when all donor molecules are bound. This asymmetry in donor- and acceptor-centric FRET measurements offers a simple and convenient strategy to deduce the stoichiometry of molecules in a bound complex. The ratio of $3^3$-FRET efficiency to the E-FRET efficiency yields the ratio ($v$) of the number of donors to the number of acceptors in the bound complex (Fig. 1c,d).

$$v = \frac{E_{A,max}}{E_{D,max}} = \frac{n_D}{n_A} \qquad (3)$$

This relation holds true so long as $E_{ij} > 0$ for at least one fluorophore pair, that is, for some $i$th donor and $j$th acceptor. Thus, this metric could, in principle, report the binding stoichiometry even when certain fluorophores are positioned beyond the Förster distance to undergo FRET ($E_{ij} \sim 0$), suggesting that this method could be apt for probing large signalling complexes.

**Functional validation using fluorescent protein concatemers.** To experimentally validate this theoretical principle, we first constructed various concatemers of ECFP and EYFP with pre-determined stoichiometries (Fig. 2a). Since the fluorophores are genetically fused to each other, all donor and acceptor molecules are assumed to be bound and the average $3^3$-FRET efficiency (Fig. 2a, black bars) and E-FRET efficiency (Fig. 2a, red bars) are determined for each concatemer from 5 to 10 transiently transfected HEK293 cells. As expected, for an ECFP-EYFP dimer, the average $3^3$-FRET efficiency is approximately equal to the average E-FRET efficiency (Fig. 2a, CY$_A$). Even though short-ening the linker between ECFP and EYFP results in enhanced $3^3$-FRET and E-FRET efficiencies, the two values reassuringly remain equal to each other (Fig. 2a, CY$_B$). For multimers with one donor and two acceptor molecules, the $3^3$-FRET efficiency is $\sim 50\%$ of the E-FRET efficiency irrespective of the relative arrangement of the fluorophores (Fig. 2a, CYY and YCY). By contrast, for multimers with two donors and one acceptor, the $3^3$-FRET efficiency is nearly twice that of the E-FRET efficiency (Fig. 2a, CYC and CCY). This general trend is confirmed for other higher order multimers (Fig. 2a, CYYY and CCCY). As such, plotting the ratio $v = E_{A,max}/E_{D,max}$ versus the known ratio of donors to acceptors ($n_D/n_A$) for each concatemer followed the identity relationship (Fig. 2b). This strong correlation

corroborates our theoretical principle and highlights the experimental feasibility and robustness of this assay.

**Stoichiometry of CaM interaction with Myosin Va.** Thus assured, we next used this method to interrogate a biologically relevant multimeric binding interaction. The unconventional myosin Va motor protein monomer is composed of an actin binding head domain and a cargo binding tail segment linked by a neck region containing six tandem IQ motifs[37]. Structurally, these tandem IQ motifs form a long contiguous helix that binds up to six CaM molecules under basal cellular Ca$^{2+}$ conditions[37]. We evaluated the stoichiometry of CaM interaction with six truncations of the myosin Va neck domain containing variable number of tandem IQ motifs. We coexpressed ECFP fused CaM with EYFP tagged myosin Va peptide containing a single IQ domain (IQ$_6$) (Fig. 3a) and quantified $3^3$-FRET and E-FRET efficiencies (Fig. 3b). Unlike for concatemers where the donor and acceptor molecules are genetically fused, the two FRET pairs are free to interact with each other depending on the concentration of the two proteins in cells. The stochastic expression of the FRET pairs in HEK293 cells allows us to obtain saturating binding isotherms for $3^3$-FRET and E-FRET efficiencies as shown in Fig. 3b. The $3^3$-FRET efficiency (Fig. 3b, left subpanel; $E_A$) is plotted against $D_{free}$, the free concentration of donors (ECFP-tagged CaM). Similarly, the E-FRET efficiency (Fig. 3b, right subpanel; $E_D$) is plotted as a function of the free concentration of acceptors (EYFP-tagged IQ$_6$ peptide). To deduce stoichiometry, we compare the saturating values of $3^3$-FRET efficiency ($E_A$,max) obtained from the subpopulation of cells where nearly all acceptors ($> 95\%$) are bound with the saturating values of E-FRET efficiency ($E_{D,max}$) obtained from the subpopulation of cells where nearly all donors ($> 95\%$) are bound. Fitting with 1:1 stoichiometry of interaction, the maximal $E_{A,max}$ ($0.164 \pm 0.003$, mean $\pm$ s.e.m., $n = 17$) is approximately equal to $E_{D,max}$ ($0.169 \pm 0.003$, mean $\pm$ s.e.m., $n = 6$) yielding a stoichiometry ratio, $v = 0.97 \pm 0.03$. By contrast, if we consider FRET between CFP-tagged CaM and YFP-tagged full length myosin Va neck domain containing all six tandem IQ motifs (Fig. 3c,d), $E_{A,max}$ ($0.255 \pm 0.005$, mean $\pm$ s.e.m.; $n = 26$; Fig. 3d, left subpanel) is substantially larger than $E_{D,max}$ ($0.043 \pm 0.007$, mean $\pm$ s.e.m.; $n = 44$; Fig. 3d, right subpanel). These efficiencies yield a stoichiometry ratio, $v = 5.91 \pm 0.15$ (mean $\pm$ s.e.m.),

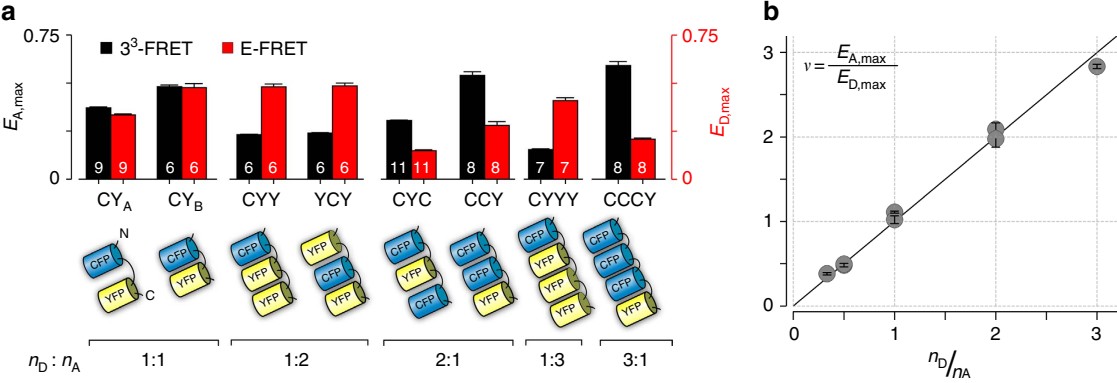

**Figure 2 | Experimental validation for FRET-based stoichiometry assay.** (**a**) Bars depict average $3^3$-FRET (black) and E-FRET (red) efficiencies for various ECFP-EYFP concatemers as described in cartoon below (mean ± s.e.m.; $n$, number of cells, as labelled for each bar). Note that the average $3^3$-FRET and E-FRET efficiencies are equal for ECFP-EYFP dimers but different for multimers. (**b**) The experimentally determined stoichiometry ratio, $v = E_{A,max}/E_{D,max}$ for each concatemer (grey symbol; mean ± s.e.m.) follows the identity relation (black fit) with expected number of donor and acceptor molecules in the complex ($n_D/n_A$) confirming the theoretical relation equation 3.

arguing that a total of six CaM molecules interact with a single full length myosin Va neck domain, consistent with available atomic structures[37]. Similar analysis of intermediate peptides containing two, three, four and five IQ motifs fused to EYFP with ECFP tagged CaM yield $E_{A,max}$ values that are approximately two-, three-, four- and five-fold larger than $E_{D,max}$ (Fig. 3e; Supplementary Fig. 4) suggesting that multiple donors interact with each peptide. More quantitatively, plotting the stoichiometry ratio ($v$) as a function of the number of IQ motifs yielded the identity relationship (Fig. 3f). These results conform well to a scheme where a single IQ motif interacts with a single CaM. In addition, the FRET-binding assays revealed relative dissociation constants ($K_{d,EFF}$) of each IQ truncation to be 800 $D_{free}$ units, equivalent to an affinity of ~25 nM (ref. 38) within range of *in vitro* estimates[39] (Supplementary Table 1). To further evaluate the robustness of our assay, we also assessed $3^3$-FRET and E-FRET efficiencies for YFP-tagged CaM and several CFP-fused myosin Va tandem IQ peptides (Supplementary Fig. 5a,c,e). With this new fluorophore arrangement, binding of multiple YFP-tagged CaM to a CFP-fused myosin Va tandem IQ peptide now yields $E_{A,max}$ that is lower than the $E_{D,max}$ as expected with the binding of multiple acceptor molecules (Supplementary Fig. 5). Reassuringly, experimentally determined stoichiometry ratio ($v$) still followed the identity relationship with expected number of donor to acceptor molecules (Supplementary Fig. 5h). In all, these results demonstrate the strong correlation between the experimentally determined stoichiometry ratio ($v$) to the number of donor to acceptor molecules in the bound complex. This outcome further corroborates the reliability and the flexibility of our FRET-based assay to determine the stoichiometry of binding interactions within live cells.

**Stoichiometry of CaM interaction with $Ca_V$ and $Na_V$ channels.** Encouraged by our ability to discriminate multimeric binding interactions, we next turned to evaluate the stoichiometry of CaM association with the voltage-gated $Ca_V$ and $Na_V$ channel complexes. Importantly, both $Ca^{2+}$-free and $Ca^{2+}$-saturated CaM can bind to each channel family to elicit various regulatory functions[7,8]. Determining the number of CaM molecules that interact with holochannel complexes *in situ* under both basal and elevated $Ca^{2+}$ conditions would help resolve a long-standing impasse in outlining the mechanistic basis of CaM regulation of these two channel families.

We first examined CaM binding to $Ca_V1.2$, a prototypic L-type channel that conveys $Ca^{2+}$ influx into diverse physiological

settings including cardiac myocytes and various neuronal cells[20,40]. To deduce stoichiometry, we evaluated $3^3$-FRET and E-FRET efficiencies between CFP-fused CaM with YFP-tagged $Ca_V1.2$ $\alpha_1$ pore-forming subunit co-expressed with other essential auxiliary components including $\beta_{2a}$ and $\alpha_2\delta$ subunits (Fig. 4a). Under resting cellular $Ca^{2+}$ conditions, plotting $3^3$-FRET efficiency ($E_A$) versus the concentration of free donor molecules ($D_{free}$) revealed a saturating binding relation as previously reported (Supplementary Fig. 6a,b; Fig. 4b). Similarly, E-FRET efficiency ($E_D$) also followed a saturating binding isotherm against the concentration of free acceptor molecules ($A_{free}$) (Supplementary Fig. 6c). Remarkably, under these conditions $E_{A,max}$ approximately equalled to $E_{D,max}$. Of note, CFP-fused CaM does not associate with membrane tethered EYFP (Supplementary Fig. 6d,e). Computing the stoichiometry ratio, $v = 1.1 ± 0.05$, (mean ± s.e.m.) demonstrates the binding of a single apoCaM to the L-type $Ca_V$ channel complex (Fig. 4c; grey bar). By contrast, upon elevating the cytosolic $Ca^{2+}$ by application of ionomycin, $E_{A,max}$ is now roughly twice $E_{D,max}$ yielding a stoichiometry ratio $v = 1.94 ± 0.14$ (mean ± s.e.m.) consistent with two $Ca^{2+}/CaM$ molecules interacting with the L-type $Ca_V$ channel complex (Fig. 4c; black bar). Moreover, these experiments also reveal that apoCaM associates with the holo-$Ca_V$ channel with a relative dissociation constant of 3,500 $D_{free}$ units (~115 nM), while $Ca^{2+}$-bound CaM binds with a substantially enhanced affinity of 700 $D_{free}$ units (~22 nM). While there are no current estimates of CaM binding affinity to holo-$Ca_V1.2$ channels[40], these findings follow trends in *in vitro* affinity measurements of key CaM-binding segments[41,42] (Supplementary Table 1). The findings here reveal a novel $Ca^{2+}$-dependent switch in the stoichiometry of CaM binding to L-type channel complex whereby a single apoCaM preassociates with the channel but a second CaM is recruited following cytosolic $Ca^{2+}$ influx (Supplementary Fig. 6f).

As with voltage-gated $Ca_V$ channels, $Na_V$ channels are also subject to potent feedback regulation by CaM with crucial implications for skeletal and cardiac muscle functions[8,43]. To evaluate stoichiometry of CaM interaction, we here probed the binding of ECFP-tagged CaM to the skeletal muscle $Na_V1.4$ channels with EYFP fused to its carboxy terminus (Fig. 4d). Similar to the L-type $Ca_V$ channels, under resting $Ca^{2+}$ conditions, $E_{A,max}$ was approximately equal to $E_{D,max}$ (Fig. 4e). The complete binding isotherms obtained using $3^3$-FRET and E-FRET methods are shown in Supplementary Fig. 7a,b. Computing the stoichiometry ratio, $v = 1.14 ± 0.07$ (mean ± s.e.m.)

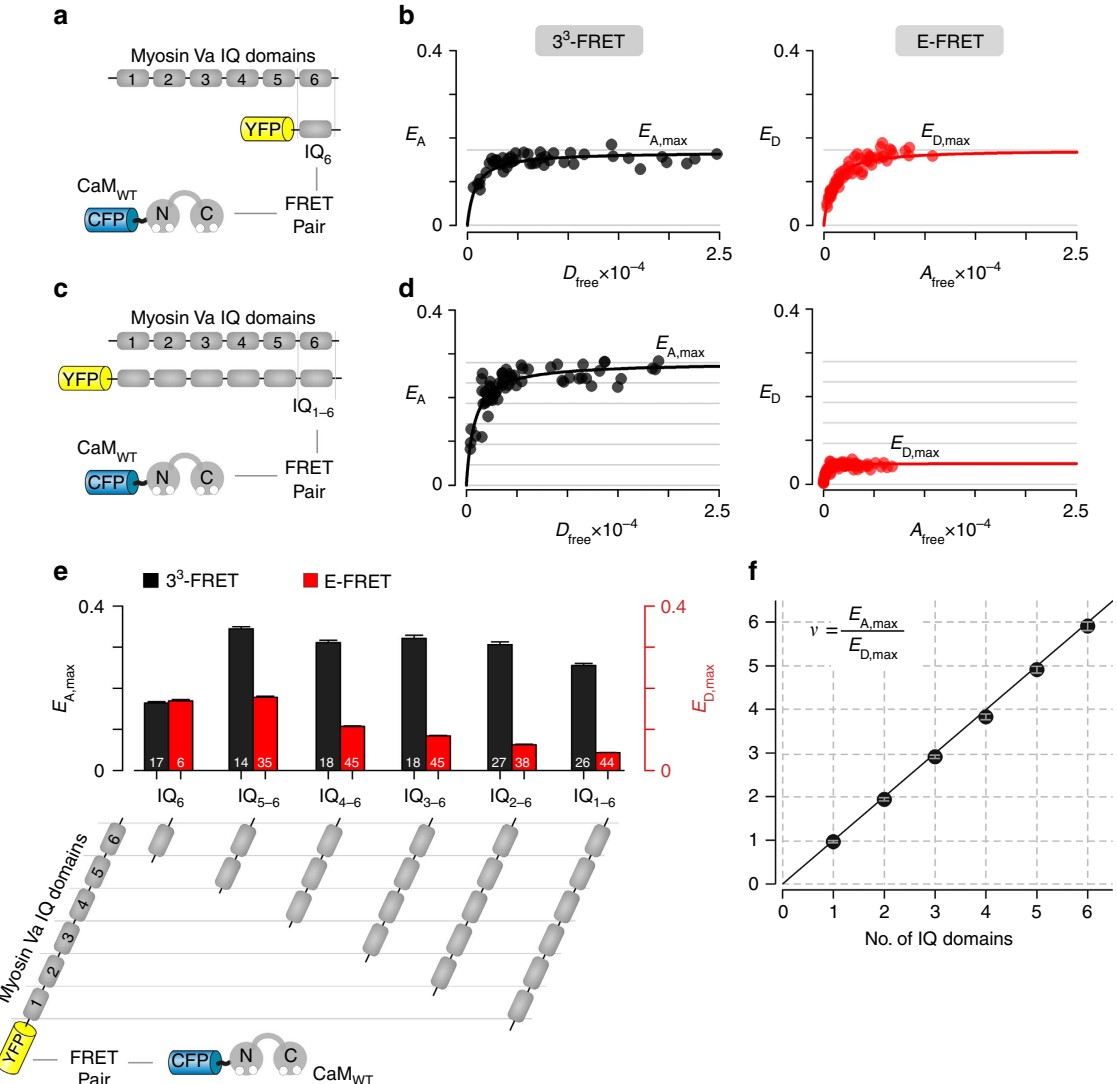

**Figure 3 | Stoichiometry of calmodulin interaction with myosin Va neck domain. (a)** Schematic illustrates FRET binding pairs ECFP tagged CaM and EYFP tagged myosin Va peptide containing a single IQ domain ($IQ_6$). **(b)** Left, $3^3$-FRET efficiency ($E_A$) is plotted against estimated free donor concentration ($D_{free}$). Each black symbol corresponds to data from a single cell. Right, E-FRET efficiency ($E_D$) is plotted as a function of estimated free acceptor concentration ($A_{free}$). The maximal $3^3$-FRET efficiency ($E_{A,max}$) is approximately equal to maximal E-FRET efficiency ($E_{D,max}$). **(c)** Cartoon illustrates FRET pairs ECFP-CaM and EYFP-tagged full length myosin Va neck domain peptide containing six IQ domains ($IQ_{1-6}$). **(d)** Left, $3^3$-FRET efficiency as a function of free donor concentration ($D_{free}$). Right, E-FRET efficiency is plotted against free acceptor concentration ($A_{free}$). In both cases, each symbol corresponds to FRET measurement from a single cell. Notice that $E_{A,max}$ is ~6-fold larger than $E_{D,max}$ suggesting 6:1 donor:acceptor stoichiometry. **(e)** Bar-graph summary depicts maximal $3^3$-FRET (black) and E-FRET (red) efficiencies for binding of ECFP-CaM with various YFP-tagged truncations of myosin Va neck domain containing varying number of IQ domains as shown in cartoon below (mean ± s.e.m.; $n$, number of cells, as labelled for each bar). **(f)** Experimentally determined FRET-based stoichiometry ratio ($v$) follow the identity relation with the number of IQ domains within each truncation shown in **e**. Each symbol, mean ± s.e.m.

demonstrates that one apoCaM associates with a single $Na_V$ channel holomolecule (Fig. 4f). Next, we considered CaM—$Na_V$ channel interaction under elevated cytosolic $Ca^{2+}$ conditions by applying ionomycin. Unlike with L-type $Ca_V$ channels, for $Na_V1.4$ $E_{A,max}$ remained approximately equal to $E_{D,max}$ (Fig. 4e), consistent with a single $Ca^{2+}$-bound CaM interacting with the $Na_V$ channel complex ($v = 1.09 ± 0.09$; mean ± s.e.m.). The binding isotherms for $Ca^{2+}$/CaM interaction with $Na_V1.4$ channels are shown in Supplementary Fig. 7c,d. With regards to relative affinities, like $Ca_V1.2$, $Na_V1.4$ binds to $Ca^{2+}$/CaM with a substantially higher affinity (150 $D_{free}$ ~ 5 nM) in comparison to $Ca^{2+}$-free CaM (1,200 $D_{free}$ ~ 39 nM) (Supplementary Table 1). These results suggest that for $Na_V$ channels, a single apoCaM preassociates to the channel complex and a single CaM remains

bound even after $Ca^{2+}$ binding to CaM. Further statistical analysis confirms CaM stoichiometry relations for both $Ca_V$ and $Na_V$ channels (Supplementary Fig. 8; Supplementary Note 2). In all, our FRET-based assay provides novel insights into a long-standing controversy in the stoichiometry of CaM interaction with voltage-gated ion channels[7,8,20]. Our findings illustrate the suitability and resolving power of our assay to discern dynamic changes in stoichiometry of signalling molecules within large macromolecular complexes such as ion channels.

## Discussion

In recent years, the use of FRET to interrogate biological molecules has been broad and rapidly expanding[26–28]. In this

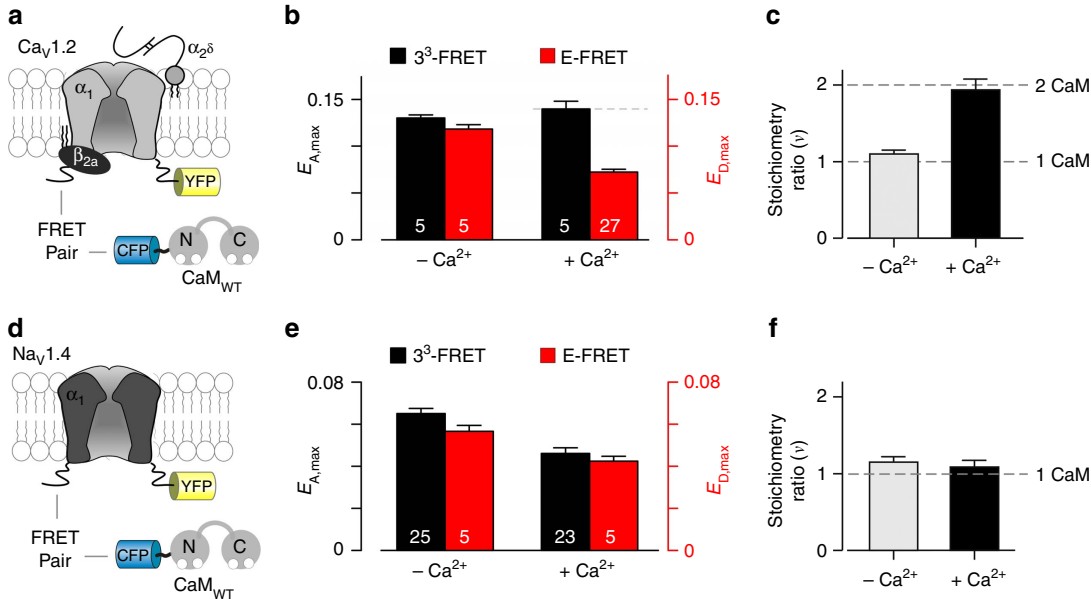

**Figure 4 | Stoichiometry of calmodulin binding to $Ca_V$ and $Na_V$ channels.** (**a**) Cartoon illustrates FRET pairs ECFP-CaM and $Ca_V1.2$ holochannel tagged with EYFP on its carboxy-terminus. The $Ca_V$ channel auxiliary subunits $\beta_{2A}$ and $\alpha_2\delta$ subunits are coexpressed. (**b**) Bar-graph summary of maximal $3^3$-FRET ($E_{A,max}$; black) and E-FRET ($E_{D,max}$; red) efficiencies under basal (− $Ca^{2+}$) and elevated $Ca^{2+}$ conditions for CaM binding to $Ca_V1.2$ (mean ± s.e.m.; $n$, number of cells, as labelled for each bar). $E_{A,max}$ and $E_{D,max}$ are approximately equal under resting $Ca^{2+}$ conditions. With high cytosolic $Ca^{2+}$ levels, $E_{A,max} \sim 2\times$ higher than $E_{D,max}$. (**c**) Computing stoichiometry ratio ($\nu$) shows that a single CaM binds to the holo-$Ca^{2+}$ channels under low $Ca^{2+}$ conditions while two CaM interact with the channel complex upon $Ca^{2+}$ elevation (mean ± s.e.m.). (**d**) Cartoon depicts FRET pairs ECFP-CaM and $Na_V1.4$ with EYFP fused to its carboxy-terminus. (**e**) Bar-graph summarizes maximal $3^3$-FRET ($E_{A,max}$; black) and E-FRET ($E_{D,max}$; red) efficiencies for CaM binding to $Na_V1.4$ (mean ± s.e.m.; $n$, number of cells, as labelled for each bar). Notice that maximal $3^3$-FRET and E-FRET efficiencies are approximately equal to each when measured under both basal and elevated $Ca^{2+}$ conditions. (**f**) Experimentally determined stoichiometry ratio ($\nu$) shows that a single CaM interacts with $Na_V$ channel complex under all $Ca^{2+}$ conditions. Format as in **c**.

regard, the FRET 2-hybrid assay has been used to quantify *in situ* strength of binding for diverse biomolecular interactions[27], ranging from transmembrane proteins such as ion channels[9,22] to cytosolic proteins crucial for cellular function[31]. For such analysis, either acceptor or donor-centric metrics of FRET efficiencies are determined from single cells and subsequently correlated with the free concentration of either donors or acceptors to determine a relative dissociation constant[9,30]. Our theoretical analysis exploits a fundamental asymmetry in these measurements to compute the stoichiometry of the bound complex as the ratio of acceptor-centric and donor-centric measurements of FRET efficiencies. Extending this analysis for binding interactions, the stoichiometry of the bound complex can be obtained as the ratio of maximal acceptor-centric measurement of FRET efficiency attained when all acceptors molecules are bound, and the maximal donor-centric measurement of FRET efficiency achieved when all donors are bound. Complementary experimental analysis using various donor-acceptor concatemers (Fig. 2) and systematic characterization of CaM binding to the myosin Va IQ domain corroborates the validity and further highlights the resolving limits of our assay (Fig. 3).

Prior attempts to discern the stoichiometry of complexes from live cells have exploited various super-resolution or single molecule approaches[44,45] and indirect methods using FRET[46,47]. Several single-molecule approaches have assessed stoichiometry of protein complexes by counting the number of photobleaching steps for fluorescence emission from a single complex[39], by assessing stochasticity in fluorescence emission using continuous-time aggregated Markov models[45] or other statistical approaches that assess brightness of single fluorophores[48,49], and by using fluorescence measurements to virtually classifying complexes

according to their conformation and stoichiometries[50]. Even so, applications of these methods to study large complexes such as pentamers[51,52] and hexamers[53,54] have been controversial with limited signal to noise ratio[50] or the presence of immature fluorophores. Moreover, these approaches often require immobilized fluorophore-tagged proteins expressed at low concentrations[44,55] or *in vitro* purified and fluorophore conjugated proteins[50] posing key technical challenges to studying the binding of small freely diffusing cytosolic proteins such as CaM to large transmembrane channel complexes[55]. In addition, robust statistical analysis of single molecule fluorescence for brightness analysis is highly sensitive to intensity of excitation light, various microscopy settings, photobleaching and motion of the cell[48]. Similarly, attempts to resolve stoichiometry using FRET have also been largely limited[49] and narrow in their generalizability. Some studies have used apparent FRET efficiencies stoiciometrically to estimate molar ratios of donors and acceptors in a single pixel, though these studies have assumed a 1:1 interaction stoichiometry and do not consider the possibility of multimeric bound complexes nor interpret the maximal FRET efficiencies[30,32,35,56]. Other approaches indirectly infer stoichiometry by exploiting *a priori* structural knowledge of the bound complex and utilized FRET to test whether individual binding partners could self-assemble or interact with each other[46,47]. Although useful in certain cases, such approaches are difficult to generalize to larger molecular complexes with limited structural information and no prior symmetry constraints. In addition, histograms of spatially resolved apparent donor-centric measurement of FRET efficiencies from single cells have been used to deduce the most likely spatial arrangement of fluorophores in the bound complex to infer stoichiometry[36,57]. Such analysis, however, is prone to ambiguities without

knowledge of actual pairwise FRET efficiencies between individual donors and acceptors in the complex[36,57].

By contrast, our present formulation holds distinct advantages. First, by computing the ratio of acceptor-centric and donor-centric measurement of FRET efficiencies, our assay directly estimates the ratio of the number of donors to acceptors in the bound complex. Second, unlike super-resolution approaches, fluorescence measurements for our method are obtained from freely diffusing complexes, and determined across a broad expression profile of donor- and acceptor-tagged binding partners. Accordingly, our assay is only minimally sensitive to unlabelled endogenous proteins and errors introduced by variable cellular expression of relevant binding partners, thereby permitting the study of molecules like CaM that are ubiquitous in all eukaryotic cells[58]. Third, our method does not require a priori knowledge of the spatial arrangement of individual donor-acceptor pairs. While both donor-centric and acceptor-centric FRET metrics incorporate the individual pairwise efficiencies of energy transfer ($E_{i,j}$ in equations 1–2), the ratio of the two metrics ($v$, equation 3) is entirely insensitive to the pairwise efficiencies. As information pertaining to the spatial arrangement of molecules is encoded entirely as the rate of energy transfer ($k_T$), it follows that our estimated stoichiometry ratio is independent of such ambiguity. Fourth, an important corollary is that our ratio of FRET efficiencies could still reliably report stoichiometry of the complex even if there is minimal energy transfer between some but not all pairs of donors and acceptors in the complex (that is, $E_{i,j} = 0$, for some $i$th donor and $j$th acceptor). This feature of our assay is particularly convenient to study large macromolecules since the tagged fluorophores in a large complex may not be at close proximity. In practice, if individual pairwise FRET efficiencies are all very small, then estimating maximal donor- or acceptor-centric estimates of FRET efficiency is challenging and prone to noise leading to indeterminacy in defining stoichiometry relations. However, so long as one donor-acceptor pair in the bound complex undergoes significant FRET, our assay can reliably report stoichiometry. For instance, the neck domain of myosin Va forms an elongated helix $\sim 225$ Å in length with CaM molecules linearly arranged $\sim 40$ Å apart[37]. Since the Förster distance for the ECFP-EYFP pair is 49Å (ref. 59), it is unlikely that all ECFP-tagged CaM that bind to the EYFP-tagged myosin Va neck domain could undergo FRET. Yet, our assay correctly identified six CaM molecules bound to the myosin Va neck domain. Finally, in terms of practicality, our approach is easy to implement requiring only a conventional epifluorescence microscope and a photomultiplier tube.

Even so, the measurement of maximal apparent FRET efficiencies may be confounded by two factors[35,60]: (1) the presence of endogenous protein and (2) incomplete maturation of fluorophores in the bound complex. As FRET 2-hybrid assay is conducted in live cells, endogenous proteins that are unlabelled may compete with their fluorescent protein tagged counterparts resulting in diminished $E_A$ and $E_D$ measurements. This reduction in apparent FRET efficiencies can be minimized if the tagged molecules are in over-abundance relative to the endogenous species. In fact, to determine stoichiometry, we overexpress the CFP- or YFP-tagged binding partners to obtain the saturating values $E_{A,max}$ and $E_{D,max}$. Under these conditions, the effect of endogenous protein is minimal[58]. A second confounding factor for determination of maximal apparent FRET efficiencies is slow or incomplete maturation of many fluorescent proteins that yield molecules with little fluorescence output[35,61]. With partial maturation of donors, the acceptor-centric measurement of FRET efficiency is diminished though the donor-centric measurement is unaffected[35] (Supplementary Note 3; Supplementary Fig. 9). Likewise with incomplete maturation of acceptors, the

donor-centric efficiency is reduced while the acceptor-centric metric is largely spared[35] (Supplementary Note 3; Supplementary Fig. 9). This effect renders the stoichiometry ratio $v$ to be also sensitive to the ratio of fractional maturation of donors and acceptors resulting in a biased estimate of actual interaction stoichiometry. That is,

$$v = \left( \frac{f_{m,D}}{f_{m,A}} \right) \cdot \frac{n_D}{n_A} = \rho \cdot \frac{n_D}{n_A} \qquad (4)$$

where $f_{m,D}$ is the fractional maturation of donors in a cell, $f_{m,A}$ is the fractional maturation of acceptors, and $\rho$ denotes the ratio $f_{m,D}/f_{m,A}$ or the bias in stoichiometry measurement due to be immature fluorophores (Supplementary Note 3). To ascertain an experimental estimate of $\rho$ for the ECFP/EYFP pair, we utilized various CFP-YFP dimers where the fluorescent proteins are genetically tethered to each other at a 1:1 stoichiometry (Fig. 1a; Supplementary Note 3). On average based on three distinct CFP-YFP dimers, we determined $\rho = 1.026$ suggesting that the mean bias in stoichiometry measurement due to incomplete maturation is less than 3%. More generally, simulations show that the ratio $v$ can reliably discern interactions with stoichiometry of 1:1–1:6 or 6:1 so long as the given FRET pair has a maturation efficiency of 90% or greater (Supplementary Fig. 9). In all, we believe this assay is well-suited to study diverse biomolecular complexes whose stoichiometry remains controversial[6,62], all within their native signalling environments. As our mathematical formulation holds true for single complexes, it is possible that extension of our assay using single-molecule FRET may complement and enrich current super-resolution approaches[44,45].

Our new findings also bear insight into the mechanism of CaM regulation of voltage-gated ion channels. The stoichiometry of CaM binding to the L-type channel has been debated for over a decade[7] with functional studies arguing for a single CaM[17] critical for modulation while structural and biochemical studies arguing for multiple CaM interacting with the channel[18–20]. Our experiments reveal an unexpected $Ca^{2+}$-dependent switch in the stoichiometry of CaM for the L-type channels—the channel appears to bind a single $Ca^{2+}$-free CaM but can bind two $Ca^{2+}$-bound CaM. One possible resolution with functional studies relate to the high intracellular $Ca^{2+}$ buffering conditions often used when probing $Ca^{2+}$-modulation of L-type channels. Under these conditions, $Ca^{2+}$-elevations are temporally brief and spatially restricted to the nanodomain of the L-type channel. Accordingly, CaM is most likely to be in its $Ca^{2+}$-free form with brief interconversion to the $Ca^{2+}$-bound form only upon channel opening[63], implying that these functional studies in actuality probe the stoichiometry of apoCaM on the channel complex. Moreover, a recent study also demonstrated that apoCaM binding itself augmented the baseline open probability of the L-type channel, and $Ca^{2+}$-modulation is a simple reversal of this effect—a model consistent with the stoichiometry of apoCaM being the relevant parameter for $Ca^{2+}$ channel modulation[64]. Nonetheless, deducing the functional role of the second CaM represents an exciting new challenge for $Ca_V$ channel biology. One possibility is that the recruited $Ca^{2+}$/CaM may enable 'functional coupling' of $Ca_V1.2$ (ref. 65). Intriguingly, this study showed that two CaM-dependent $Ca_V1.2$ functions—the canonical $Ca^{2+}$-dependent inactivation and 'coupled channel gating'—exhibited distinct sensitivities to a CaM inhibitory peptide suggesting that the two functions maybe mediated by two unique CaM[65]. Given that we detect the binding of two $Ca^{2+}$/CaM, our findings lend further support to this hypothesis. It is also possible that the second CaM may serve functions beyond channel gating such as channel trafficking[13,14], signalling to other enzymes[16], or could be shuttled to the nucleus following

$Ca^{2+}$-activation to trigger gene-transcription[15]. For voltage-gated $Na_V$ channels, the CaM regulatory scheme appears to be simpler, involving a single CaM prebound to the channel complex that interconverts between a $Ca^{2+}$-free and $Ca^{2+}$-bound form—a scheme that is consistent with functional studies[22,43]. Although the $Na_V$ channel cytosolic domains feature multiple CaM binding sites, the functional relevance of these sites is yet to be fully substantiated[8,43]. Finally, mutations in CaM genes have been associated with multiple forms of life-threatening cardiac arrhythmias[66]. Given the prominent role of $Ca_V$ and $Na_V$ channels in the electrical stability of the heart, the distinct stoichiometric modes of CaM binding to these channels may hold pathological consequences that our method could help resolve. Altogether these findings exemplify the utility of our assay in addressing outstanding questions while generating hypotheses to explore new frontiers.

Our assay represents a valuable general strategy to evaluate the stoichiometry of large macromolecular complexes and to probe dynamic changes in stoichiometry associated with cellular signalling events. Overall, this method enriches the current repertoire of tools available to pursue quantitative biochemistry within the realm of living cells.

## Methods

**Molecular biology.** All ECFP-EYFP concatemers were constructed from ECFP and EYFP clones[9,67] as described for each clone. Construct $CY_A$ (Fig. 2a) was constructed by fusing EYFP to the carboxy-terminus of ECFP using the linker with the protein sequence 'SRAQASNSAVDGTAGPGSIAT'. For construct CYC, an EYFP molecule is interposed between two ECFP molecules utilizing the linker 'SGSSSGSSSLAGIEGRSSSGSSSGS'. By contrast, the YCY construct contains two EYFP molecules bookending an ECFP molecule using the same linker 'SGSSSGSSSLAGIEGRSSSGSSSGS'. To construct other concatemers shown in Fig. 2, we engineered the 5′ end of ECFP and EYFP to contain unique restriction site EcoRI followed by a Kozak consensus start sequence followed by the unique restriction site SpeI and 3′ end to contain the unique restriction site XbaI followed by stop codon ('TAA') terminated by unique restriction site ApaI thus yielding two constructs ECFP-pCDNA3 and EYFP-pcDNA3. To construct $CY_B$ dimer, we digested EYFP-pCDNA3 with SpeI and ApaI and ligated into ECFP$^+$-pcDNA3 that was digested with XbaI and ApaI. This manoeuvre resulted an ECFP-EYFP dimer fused with a two residue linker 'SS'. To generate the CYY trimer, we digested EYFP-pCDNA3 with SpeI and ApaI and ligated into the $CY_B$ dimer vector digested with XbaI and ApaI resulting in the two residue linker 'SS' adjoining each EYFP. To generate the CYYY tetramer, we followed the same strategy by digesting EYFP-pCDNA3 with restriction enzymes SpeI and ApaI but now ligating this insert into the CYY trimer vector that was digested using XbaI and ApaI. To generate trimer CCY, we digested $CY_B$ dimer with SpeI and ApaI, and ligated into ECFP-pcDNA3 vector digested using XbaI and ApaI. Finally, to construct the CCCY tetramer, we digested the CCY trimer with SpeI and ApaI and ligated into the ECFP-pcDNA3 vector digested using XbaI and ApaI. In all cases, the size of the entire concatemer construct was tested by restriction digest using EcoRI and KpnI. In addition, terminal ECFP and EYFP molecules in each construct were also sequence verified.

ECFP-tagged CaM and EYFP-tagged CaM were made as described[9]. For EYFP and ECFP tagged truncations of mouse myosin Va[68] neck domain (Uniprot Q99104), we replaced CaM with the relevant truncation of PCR-amplified myosin Va neck domain using unique restriction sites NotI and XbaI that flank CaM[9]. Importantly, all such constructs contain a three amino-acid linker 'AAA' that joins ECFP or EFYP with various truncated myosin Va tandem IQ peptides. The full length neck domain peptide containing six tandem IQ domains (IQ1-6) is composed of residues L[766]-K[920], the five tandem IQ domain (IQ$_{2-6}$) truncation is composed of residues M[789]-K[920], the four tandem IQ domain (IQ$_{3-6}$) is composed of residues R[814]-K[920], the three tandem IQ domain (IQ$_{4-6}$) is composed of residues R[837]-K[920], the two tandem IQ domain, IQ$_{5-6}$, construct contains residues R[862]-K[920]; the single IQ domain, IQ$_6$ construct contains residues T[885]-K[920]. The $Ca_V1.2$-EYFP construct was engineered by fusing EYFP to the carboxy-terminus of $\alpha_{1C}$ channel subunit as previously described[9]. To construct $Na_V1.4$-YFP, we first removed the stop codon from the full-length rat $Na_V1.4$ pCDNA3 (ref. 43) following PCR amplification of the carboxy-terminal $\sim$870 bp segment and ligation using unique restriction sites KpnI and XbaI. Subsequently, the EYFP gene was inserted into the $Na_V$ channel carboxy-terminus following PCR amplification and restriction digest using enzymes SpeI/XbaI. The resultant plasmid contains an EYFP fusion construct with linker 'SS' joining the terminal 'SLV' residues of $Na_V1.4$ with the initial segment of EYFP (residues 'VSKG'). For all constructs, PCR amplified segments were verified by sequencing. Expression of all constructs in mammalian expression systems was driven by the CMV promoter.

**Transfection of HEK293 cells.** HEK293 cells (ATCC) CRL1573 were cultured on glass-bottom dishes and transfected with polyethylenimine (PEI) 25 kDa linear polymer (Polysciences #2396602), before epifluorescence microscope imaging. Briefly, in a sterile tube the relevant plasmid DNA are mixed together in 200 µl of serum-free DMEM media. PEI is added to each sterile tube at 3:1 ratio of PEI (µg) to total DNA (µg). The mixture was incubated at room temperature for 10 min before adding to cells. FRET experiments were performed at room temperature 1–3 days post-transfection. For live-cell binding assays, we typically co-transfected a variety of ratios of plasmids encoding CFP-tagged and YFP-tagged binding partners to robustly resolve saturating values of apparent FRET efficiencies. For example, to resolve $E_{A,max}$, we co-transfected 1 µg of plasmid encoding YFP-tagged binding partner to 2–3 µg of plasmid encoding the CFP-tagged partner. On the other hand, to resolve $E_{D,max}$, we co-transfected 2–3 µg of plasmid encoding YFP-tagged binding partner to 1 µg of plasmid encoding the CFP-tagged partner. HEK293 cells were not contaminated by mycoplasma.

**FRET optical imaging.** We conducted FRET 2-hybrid experiments in HEK293 cells cultured on 35 mm glass-bottom dishes, using an inverted Nikon TE300 Eclipse ($\times 40$ (1.3 n.a.) objective) fluorescence microscope and custom fluorometer system (University of Pennsylvania Biomedical Instrumentation Group) as extensively described by our laboratory[69]. Briefly, a 150 W short-gap Xenon arc-lamp (Optiquip) gated by a computer controlled shutter was used to deliver excitation light. Epifluorescence emission from entire individual cells isolated using an image-plane pinhole was measured using an ambient temperature photomultiplier tube. Shutter control, data acquisition, automated filter-cube control and dark-current subtraction were attained using custom MATLAB (MathWorks) software. Exact specifications of CFP, YFP and FRET filter cubes are previously described[9] and also detailed in Supplementary Note 1 for convenience. Experiments utilized a bath Tyrode's solution (138 mM NaCl, 4 mM KCl, 1 mM MgCl$_2$, 10 mM HEPES (pH 7.4 using NaOH), 0.2 mM NaHPO$_4$ and 5 mM D-glucose) containing either 2 mM $Ca^{2+}$ for experiments probing apoCaM binding or 10 mM $Ca^{2+}$ with 4 µM ionomycin (Sigma-Aldrich, MO, USA) for $Ca^{2+}$/CaM-binding experiments. To remove autofluorescence and background light scatter, average fluorescence intensities from untransfected cells of similar confluency were subtracted from same-day experimental values for each filter cube. In general, fluorescence measurements were only obtained if fluorescence signal:background for FRET cube was $>6$:1. 3$^3$-FRET ($E_A$) and E-FRET ($E_D$) efficiencies were computed from CFP ($S_{CFP}$), YFP($S_{YFP}$), FRET($S_{FRET}$) cube measurements as introduced in our prior publications[9,38] and elaborated in Supplementary Note 1. E-FRET efficiency ($E_D$) measurement methodology was first developed and refined in other laboratories[29]. Spectral ratio $R_{D1}$ and $R_{D2}$ were determined from cells expressing ECFP alone, and $R_{A1}$ was determined from cells expressing EYFP alone on the same day of experimentation. For CFP-YFP concatemers, the average 3$^3$-FRET and E-FRET efficiencies are measured from several cells.

In the case of binding interactions (Figs 3 and 4), we determined FRET 2-hybrid binding curves using methods as described in previous publications[9,29,69]. For multimeric interactions involving multiple donors and a single acceptor, we assumed an independent identical binding scheme. As detailed in previous publications[9,69], CFP$_{EST}$ and YFP$_{EST}$ is proportional to the number of donors ($N_D$) and acceptors ($N_A$) given by CFP$_{EST} = N_D \cdot I_0 \cdot C$ and YFP$_{EST} = N_A \cdot I_0 \cdot C$, where $I_0$ is intensity of the excitation light and $C$ is a proportionality constant (Supplementary Note 1). Experimentally, these values can be determined as,

$$\text{CFP}_{EST} = \frac{R_{D1} \cdot S_{CFP}(DA, 440, 480)}{M_D \cdot (1 - E_D)};$$
$$\text{YFP}_{EST} = \frac{R_{A1} \cdot S_{YFP}(DA, 500, 535)}{M_A} \tag{5}$$

Here, $M_A$ and $M_D$ are instrument-specific constants corresponding to the brightness of a single EYFP and ECFP molecule measured using FRET cube[9,69]. These instrument-specific constants can be computed from *in vitro* measurements of donor (ECFP) and acceptor (EYFP) excitation and emission spectra and specific knowledge of the microscope fluorescence filters as previously described[9]. Briefly, $M_A \approx [\varepsilon_A(\lambda)]_{\lambda = 430-450 \, nm} \cdot [f_A(\lambda)]_{\lambda = 505-575 \, nm} \cdot QY_A$ and $M_D \approx [\varepsilon_D(\lambda)]_{\lambda = 430-450 \, nm} \cdot [f_D(\lambda)]_{\lambda = 505-575 \, nm} \cdot QY_D$. Here, $[\varepsilon_A(\lambda)]_{\lambda = 430-450 \, nm}$ and $[\varepsilon_D(\lambda)]_{\lambda = 430-450 \, nm}$ are average values of the molar extinction coefficients of ECFP and EFYP over the excitation bandwidth of FRET filter cube. $[f_D(\lambda)]_{\lambda = 505-575 \, nm}$ and $[f_A(\lambda)]_{\lambda = 505-575 \, nm}$ are average values of the emission spectra for ECFP and EYFP over the emission bandwidth for the FRET filter cube. Importantly, the both $f_A$ and $f_D$ spectra are normalized such that the total area under each spectrum is 1.

The free donor concentration ($D_{free}$) can be estimated with the dissociation constant $K_{d,EFF}$ fit iteratively through least-squares minimization[9]. For each cell,

$$D_{free} = \frac{-\left( \text{YFP}_{EST} \cdot (n_D/n_A) + K_{d,EFF} - \text{CFP}_{EST} \right)}{2}$$
$$+ \frac{\sqrt{\left( \text{YFP}_{EST} \cdot (n_D/n_A) + K_{d,EFF} - \text{CFP}_{EST} \right)^2 + 4 \cdot \text{CFP}_{EST} \cdot K_{d,EFF}}}{2} \tag{6}$$

where CFP$_{EST}$ is proportional to the number of ECFP molecules, YFP$_{EST}$ is proportional to the number of EYFP molecules, and $K_{d,EFF}$ is the effective dissociation constant. Once $D_{free}$ is estimated, $A_{free}$ is determined as

$YFP_{EST} - (CFP_{EST} - D_{free})/(n_D/n_A)$. Given this scheme, the $3^3$-FRET efficiency can be shown to be,

$$E_A = E_{A,FIT} \frac{D_{free}}{D_{free} + K_{d,EFF}} \quad (7)$$

$E_{A,FIT}$ here corresponds to the maximal $3^3$-FRET efficiency. No corresponding closed form relation exists for E-FRET efficiency since different cells have different amounts of total donors and acceptors; nonetheless, an approximate relation can be derived as follows:

$$E_D = E_{D,FIT} \frac{A_{free}}{A_{free} + K_{d,EFF} \cdot (n_A/n_D)} \quad (8)$$

$E_{D,FIT}$ in this case corresponds to the maximal E-FRET efficiency and $K_{d,EFF}$ is the dissociation constant from above.

By contrast for interactions with multiple acceptors and a single donor (1:$n_A$), with independent identical binding, the free concentration of acceptors ($A_{free}$) can be obtained iteratively as,

$$A_{free} = \frac{-\left(CFP_{EST} \cdot (n_A/n_D) + K_{d,EFF} - YFP_{EST}\right)}{2} + \frac{\sqrt{\left(CFP_{EST} \cdot (n_A/n_D) + K_{d,EFF} - YFP_{EST}\right)^2 + 4 \cdot YFP_{EST} \cdot K_{d,EFF}}}{2} \quad (9)$$

where $CFP_{EST}$ and $YFP_{EST}$ are total amount of donors and acceptors computed using equation 5. In this case, the E-FRET efficiency follows the relation,

$$E_D = E_{D,FIT} \frac{A_{free}}{A_{free} + K_{d,EFF}} \quad (10)$$

where $E_{D,FIT}$ corresponds to the maximal E-FRET efficiency (equation 2). For this scenario, the $3^3$-FRET efficiency approximately follows the relation,

$$E_A = E_{A,FIT} \frac{D_{free}}{D_{free} + K_{d,EFF} \cdot (n_D/n_A)} \quad (11)$$

$E_{A,FIT}$ is an estimator for the maximal $3^3$-FRET efficiency (equation 1). It is important to note that for the present analysis we do not interpret the half saturation concentrations.

To ensure that the binding curves reach saturation, we verified that the concentrations of donors and acceptors are sufficiently high that further increase in these concentrations would not change the measured $E_A$ and $E_D$ values. Thus, we ensured that the expected fractional change in $E_A$ $\left(\delta_{E_{A,max}}\right)$ as a result of doubling the maximal observed free concentration of donors ($D_{exptl-max}$) for a given experimental condition is less than 5% for all interactions probed. That is,

$$\delta_{E_{A,max}} = \frac{\Delta E_A}{E_A}\left(D_{exptl-max}\right) = \frac{E_A\left(2D_{exptl-max}\right) - E_A\left(D_{exptl-max}\right)}{E_A\left(D_{exptl-max}\right)}$$

$$= \frac{1}{E_A\left(D_{exptl-max}\right)}$$

$$\cdot \left(\left(\left.\frac{\partial E_A}{\partial D_{free}}\right|_{D_{exptl-max}}\left(D_{exptl-max}\right) + E_A\left(D_{exptl-max}\right)\right) - E_A\left(D_{exptl-max}\right)\right)$$

$$= \frac{D_{exptl-max}}{E_A\left(D_{exptl-max}\right)} \cdot \left.\frac{\partial E_A}{\partial D_{free}}\right|_{D_{exptl-max}} < 5 \%$$

Similarly for E-FRET, we verified that the maximal change in the observed maximal E-FRET efficiency is less than 5% for all binding interactions probed. That is for a given experiment if the highest concentration of free acceptors probed is $A_{exptl-max}$ then,

$$\delta_{E_{D,max}} = \frac{\Delta E_D}{E_D} = \frac{A_{exptl-max}}{E_D\left(A_{exptl-max}\right)} \cdot \left.\frac{\partial E_D}{\partial A_{free}}\right|_{A_{exptl-max}} < 5 \%$$

Since $E_A$ and $E_D$ curves are saturating curves with slopes $dE_A/dD_{free}$ and $dE_A/dA_{free} \rightarrow 0$ for large $D_{free}$ and $A_{free}$ values, this error measurement is likely an over-estimate. For all cases, $E_{A,max}$ is computed as average $3^3$-FRET efficiency of cells with $A_{free}/YFP_{EST} < 5\%$ (that is, greater than 95% acceptors are bound) and $E_{D,max}$ is measured as average E-FRET efficiency of cells with $D_{free}/CFP_{EST} < 5\%$ (that is, greater than 95% donors are bound). Both criteria were pre-established. Similar results are also obtained with more relaxed criteria (15%). We collected sufficient number of cells to obtain at least five data points to resolve $E_{A,max}$ and $E_{D,max}$. Typically, these data were collected from at least three independent transfections performed over three different days.

For all FRET efficiency measurements, spurious FRET relating to unbound ECFP and EYFP moities was subtracted[38]. For $3^3$-FRET, spurious FRET is linearly proportional to the concentration of CFP molecules, and the experimentally determined slope$A_{3^3-FRET}$ was obtained from cells coexpressing ECFP and EYFP fluorophores. For E-FRET, spurious FRET is linearly proportional to the concentration of YFP molecules also obtained from cells co-expressing ECFP and EYFP[38].

**Data availability.** The authors confirm that all relevant data are included in the paper and/or its Supplementary Information files and is available at request from the authors. MATLAB codes for simulations shown in Supplementary Fig. 8 are available upon request from the authors.

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

## Acknowledgements

This manuscript is dedicated to the memory of D.T.Y. We thank Frank Bosmans, Rahul Banerjee, Takanari Inoue, John Issa, Christopher Lingle, Po-Wei Kang, Jacqueline Niu, Christian Wahl and Gordon Tomaselli for valuable comments on our manuscript. We thank Sandy Hernandez for pointing us to unconventional myosin Va/CaM experiments. We are also grateful to M.G. Erickson for construction of ECFP-EYFP concatemers (CY$_A$, CYC and YCY). Supported by grants from the NIMH (to D.T.Y. and to M.B.-J.).

## Author contributions

M.B.-J. and D.N.Y. undertook extensive molecular biology and FRET 2-hybrid experiments. M.B.-J. furnished theoretical derivations for determining stoichiometry from FRET efficiencies. M.B.-J. and D.T.Y. conceived the project, refined experimental design and hypothesis. M.B.-J. wrote the manuscript.

## Additional information

**Competing financial interests:** The authors declare no competing financial interests.

