## [Peer Review File · Nature Communications]

Reviewers' Comments:

Reviewer #1 (Remarks to the Author)

using state-of-art technology of intermolecular FRET. Here, the research group has further expanded their approach by integrating mathematical analysis and show a critical difference between CaM-Cav channel complex and CaM-Nav complex.

Fig. 1 shows the scheme of analysis.

Fig. 2 shows the validation of the analysis by using concatamers of CFP and YFP.

Fig. 3 shows CaM binding to IQ domains of myosin Va, which validate their method.

Fig. 4 shows the authors' proposal that the mode of binding to CaM to the calcium ion channel Cav1.2 is different from that of the sodium ion channel Nav1.4.

This work has been well done in an organized way and describes in detail on the measurement of FRET in living cells, which will be informative to many researchers who are not familiar with FRET measurement.

B.Originality and interest: The methodology has been already described, although the authors introduced new terminology to describe their methods. The mode of CaM binding to ion channels may be new.

C.Data & methodology: Described in detail.

D.Appropriate use of statistics and treatment of uncertainties: Not described. The number of independent experiments should also be stated.

E.Conclusions: A critical negative control is missing as described below. As for the mode of binding of CaM to ion channels, the conclusion was drawn from a single experiment. The authors may need to tone down.

F. Suggested improvements are listed below:

1. Terminology: There is a strict definition on the FRET efficiency. It is misunderstanding to use the following terms, acceptor-centric FRET efficiency (EA) donor-centric FRET efficiency (ED).
2. Reproducibility should be shown more clearly. For example, in Figure 2, it is not clearly stated what n stands for: The number of cells or dishes?
3. Figure 4. A critical negative control is missing. It is mandatory to show the data of some membrane-bound proteins, which does not bind to CaM.

Reviewer #2 (Remarks to the Author)

The current manuscript by Ben-Johny et al. describe a methodology aiming to recover stoichiometry's of complexes between two different partners, making use of FRET. Previous published methods showed that for a 1:1 complex, donor- (E-FRET) and acceptor (33) centric FRET measurements as described in this paper recover identical values. However, for other stoichiometries, these two methods recover different apparent FRET efficiencies. The authors make use of the asymmetry in the results from these donor- and acceptor centric FRET measurements and show how the stoichiometry in hetero-oligomeric complexes can be derived from the maximal FRET efficiencies (obtained at saturating donor or acceptor concentrations) obtained from each measurement strategy. The authors demonstrate the reliability of this method through its application to a well characterized system (CaM-Myosin Va neck domain), where the method recovered the expected stoichiometries of CaM-Myosin Va neck domain interaction for different truncations of the former domain, independently of the choice of FRET donor and acceptor labeling for each of the constructs. Importantly, this methodology allowed for the characterization of the stoichiometry of the CaM-Cav channel stoichiometry for interaction both at basal and elevated calcium levels, confirming that an additional CaM is expected to associated with the voltage gated calcium channel after an increase in calcium levels. Finally, the impact of donor and acceptor immaturation on the results of this methodology is quantified in detail.

The manuscript is well written and accurately describes and proves the presented methodology.

Nevertheless, I have some comments that must be addressed:

- In the supplementary materials, the authors define ED in eq. S1.8. According to this definition: ED is the ratio of quenched donor fluorescence (Fluorescence of D in the absence of FRET – fluorescence of donor in the presence of FRET) over the Donor fluorescence intensity in the absence of FRET (This is also the definition of E in Eq.2 of Chen et al. Biophys. J. 91 (2006) L39–41) . However, this FRET efficiency is not expected to scale with donor occupation numbers in the oligomer, as this is an absolute value of the fraction of donor photons lost due to FRET. The ED given by the authors in Eq. 2 of the manuscript is “the expected number of energy transfer events per donor molecule in the complex given that all such donor molecules are excited” By normalizing to the number of donors, this is no longer an absolute measurement and does not seem to correspond to the initial definition provided in Eq. S1.8. Please clarify.

- the authors should explain in greater detail the calculation of MA and MD calibration factors, as these are crucial for the application of the methodology.

- The analysis of FRET binding isotherms should allow for the recovery of dissociation constants, however these are neither presented or discussed on the basis of literature values. The authors should comment on this.

Minor Comments:

- The format of Reference List should be homogenized.

Pag 3, line 89: Some specific FRET events (as FRET events between identical molecules) might not result in a red shift of fluorescence.

Pag 4, line 109: “...quenching of the donor emission spectrum and (2) enhancement in the acceptor emission spectrum” should be replaced by “quenching of the fluorescence intensity of the donor (2) and increase in the fluorescence intensity of the acceptor”.

Page 11, line 302: Rewrite sentence: “In addition, histograms of spatially-resolved apparent donor-centric FRET efficiencies from single cells to deduce the most likely spatial arrangement of fluorophores in the bound complex to infer stoichiometry”

Supplementary materials:

Pag3 and 4: Indexes used for some of the quantities are shown with different values while apparently referring to the same quantity. Namely:

- SYFP((A,500,535) and SYFP(A,500,530LP);

- SFRET(DA,440,535LP) and SFRET(DA,440,535)

Pag 6: I0 is not identified

Reviewer #3 (Remarks to the Author)

In this study, Ben-Johny et al developed a novel strategy to use FRET signals to determine the stoichiometry of macromolecular complexes. They used the methodology to determine the stoichiometry of calmodulin binding to calcium and sodium channels. Their data show that, at resting Ca, only 1 CaM binds to CaV1.2 channels. However, when Ca increases, 2 CaMs bind to the channel. NaV channels seem to only bind to 1 CaM.

Overall the quality of the data and analyses is excellent. The new method will undoubtedly be useful to many investigators adding to hetero FRET as a means to determine protein stoichiometry.

I have a few suggestions. First, the authors should comment on the implications of the study on recent work by Dixon et al (2015) for channel-to-channel interactions, as they propose a 2CaM:1CaV1.2 channel model in which 1 CaM binds in a Ca-dependent manner to the channel. It would also be helpful if the stoichiometry of CaM/CaV1.2 complex is determined in a native cell.

RESPONSE TO REVIEWERS (NCOMMS-16-16204)

Reviewer #1, expert in FRET sensors (Remarks to the Author):

(1.1) using state-of-art technology of intermolecular FRET. Here, the research group has further expanded their approach by integrating mathematical analysis and show a critical difference between CaM-Cav channel complex and CaM-Nav complex.

Fig. 1 shows the scheme of analysis.

Fig. 2 shows the validation of the analysis by using concatamers of CFP and YFP.

Fig. 3 shows CaM binding to IQ domains of myosin Va, which validate their method.

Fig. 4 shows the authors' proposal that the mode of binding to CaM to the calcium ion channel Cav1.2 is different from that of the sodium ion channel Nav1.4.

This work has been well done in an organized way and describes in detail on the measurement of FRET in living cells, which will be informative to many researchers who are not familiar with FRET measurement.

Thank you for the highly positive assessment.

(1.2) B.Originality and interest: The methodology has been already described, although the authors introduced new terminology to describe their methods. The mode of CaM binding to ion channels may be new.

Thank you.

(1.3) C.Data & methodology: Described in detail.

Thank you

(1.4) D.Appropriate use of statistics and treatment of uncertainties: Not described. The number of independent experiments should also be stated.

The manuscript reports n the number of cells from which maximal FRET efficiency was assessed. Since the expression of fluorophore tagged proteins is independent between different cells, the FRET measurements assessed from individual cells are independent observations and are averaged as typically done for these experiments. Nonetheless, in order to obtain at least 5 data points to resolve $E_{A,max}$ AND $E_{D,max}$, we typically collected FRET data from at least three “independent” transfections conducted over three different days. This information is provided in the Methods section.

(1.5) E.Conclusions: A critical negative control is missing as described below. As for the mode of binding of CaM to ion channels, the conclusion was drawn from a single experiment. The authors may need to tone down.

The requested negative control is presented in Supplementary Fig. 6d-e. For further details see response 1.8.

F. Suggested improvements are listed below:

(1.6) 1. Terminology: There is a strict definition on the FRET efficiency. It is misunderstanding to use the following terms, acceptor-centric FRET efficiency (E_A) donor-centric FRET efficiency (E_D).

Thank you for this important suggestion. We agree. We intended to use the term acceptor-centric metric of FRET efficiency and donor-centric metric of FRET efficiency as two means to estimate true FRET efficiency. We have corrected this.

(1.7) 2. Reproducibility should be shown more clearly. For example, in Figure 2, it is not clearly stated what n stands for: The number of cells or dishes?

n describes number of cells as is customarily reported for FRET experiments (see references 9, 30, and 31). This clarification is included in the figure legend.

(1.8) 3. Figure 4. A critical negative control is missing. It is mandatory to show the data of some membrane-bound proteins, which does not bind to CaM.

Thank you for this suggestion. This data is now presented in Supplementary Fig. 6d-e. Of note, membrane tethered YFP do not associate with ECFP-tagged CaM. Thus, both E_A and E_D lie on the zero line.

Reviewer #2, expert in FRET (Remarks to the Author):

(2.1) The current manuscript by Ben-Johny et al. describe a methodology aiming to recover stoichiometry's of complexes between two different partners, making use of FRET. Previous published methods showed that for a 1:1 complex, donor- (E -FRET) and acceptor (33) centric FRET measurements as described in this paper recover identical values. However, for other stoichiometries, these two methods recover different apparent FRET efficiencies. The authors make use of the asymmetry in the results from these donor- and acceptor centric FRET measurements and show how the stoichiometry in hetero-oligomeric complexes can be derived from the maximal FRET efficiencies (obtained at saturating donor or acceptor concentrations) obtained from each measurement strategy. The authors demonstrate the reliability of this method through its application to a well characterized system (CaM-Myosin Va neck domain), where the method recovered the expected stoichiometries of CaM-Myosin Va neck domain interaction for different truncations of the former domain, independently of the choice of FRET donor and acceptor labeling for each of the constructs. Importantly, this methodology allowed for the characterization of the stoichiometry of the CaM-Cav channel stoichiometry for interaction both at basal and elevated calcium levels, confirming that an additional CaM is expected to associated with the voltage gated calcium channel after an increase in calcium levels. Finally, the impact of donor and acceptor immaturation on the results of this methodology is quantified in detail.

The manuscript is well written and accurately describes and proves the presented methodology. Nevertheless, I have some comments that must be addressed:

Thank you for the highly positive assessment.

(2.2) - In the supplementary materials, the authors define ED in eq. S1.8. According to this definition: ED is the ratio of quenched donor fluorescence (Fluorescence of D in the absence of FRET – fluorescence of donor in the presence of FRET) over the Donor fluorescence intensity in the absence of FRET (This is also the definition of E in Eq.2 of Chen et al. Biophys. J. 91 (2006) L39–41). However, this FRET efficiency is not expected to scale with donor occupation numbers in the oligomer, as this is an absolute value of the fraction of donor photons lost due to FRET. The ED given by the authors in Eq. 2 of the manuscript is “the expected number of energy transfer events per donor molecule in the complex given that all such donor molecules are excited” By normalizing to the number of donors, this is no longer an absolute measurement and does not seem to correspond to the initial definition provided in Eq. S1.8. Please clarify.

The two definitions are identical. We here furnish both formal and informal explanations demonstrating this identity.

Formal analysis. To demonstrate this equality let us reconsider derivation in Supplementary Note 1.3:

First, in the absence of FRET, the total CFP output is given by,

$$CFP_{\text{FRET|after}} = N_D \cdot I_0 \cdot G_x(D, \lambda_{\text{ex},x}) \cdot F_x(D, \lambda_{\text{em},x}) \quad (\text{R1})$$

This equation can be derived by considering the probability that a donor is in the excited state assuming there is no FRET. Here, since there is no FRET, both donor molecules that are free and those that are bound to a “photobleached” acceptor, will have identical fluorescence output.

Thus, for all donor molecules,

$$\frac{dP_{D^*}}{dt} = I_0 \cdot G_x(D, \lambda_{\text{ex},x}) \cdot P_D - (k_D + k_{D,\text{nr}}) \cdot P_{D^*}$$

As usual, if we assume steady-state and “low-excitation” limit ($P_D \sim 1$),

$$P_{D^*} = \frac{I_0 \cdot G_x(D, \lambda_{\text{ex},x})}{k_D + k_{D,\text{nr}}}$$

The total number of photons emitted from all N_D molecules is given by,

$$N_D \cdot k_D \cdot P_{D^*} \cdot \hat{F}_x(D, \lambda_{\text{em},x})$$

This relation can be simplified by recalling that the quantum yield, $QY_D = k_D / (k_D + k_{D,\text{nr}})$, and substituting P_{D^*} ,

$$N_D \cdot k_D \cdot \frac{I_0 \cdot G_x(D, \lambda_{\text{ex},x})}{k_D + k_{D,\text{nr}}} \cdot \hat{F}_x(D, \lambda_{\text{em},x}) = N_D \cdot I_0 \cdot G_x(D, \lambda_{\text{ex},x}) \cdot QY_D \cdot \hat{F}_x(D, \lambda_{\text{em},x})$$

Thus, $CFP_{\text{FRET|after}} = N_D \cdot I_0 \cdot G_x(D, \lambda_{\text{ex},x}) \cdot F_x(D, \lambda_{\text{em},x})$

Now, $CFP_{\text{FRET|before}} = CFP_x(\lambda_{\text{ex},x}, \lambda_{\text{em},x}, \text{direct})$ - the very exact entity we measure as CFP fluorescence via direct excitation with all acceptors fully intact. Thus, by Eq. S1.46,

$$CFP_{\text{FRET|before}} = N_D \cdot I_0 \cdot G_x(D, \lambda_{\text{ex},x}) \cdot F_x(D, \lambda_{\text{em},x}) - N_B \cdot I_0 \cdot G_x(D, \lambda_{\text{ex},x}) \cdot F_x(D, \lambda_{\text{em},x}) \sum_{i=1}^{n_D} \sum_{j=1}^{n_A} E_{i,j}$$

Thus, computing E_D as per Eq. S1.8 by substituting R1 and Eq. S1.46 yields,

$$E_D = \frac{CFP_{\text{FRET|after}} - CFP_{\text{FRET|before}}}{CFP_{\text{FRET|after}}} = \frac{N_B \cdot I_0 \cdot G_x(D, \lambda_{\text{ex},x}) \cdot F_x(D, \lambda_{\text{em},x}) \cdot \sum_{i=1}^{n_D} \sum_{j=1}^{n_A} E_{i,j}}{N_D \cdot I_0 \cdot G_x(D, \lambda_{\text{ex},x}) \cdot F_x(D, \lambda_{\text{em},x})} = \frac{N_B}{N_D} \sum_{i=1}^{n_D} \sum_{j=1}^{n_A} E_{i,j}$$

If all donor molecules are bound: $N_D = N_B \cdot n_D$. The maximal FRET efficiency thus reduces to main text Eq. 2:

$$E_{D,\text{max}} = \frac{1}{n_D} \sum_{i=1}^{n_D} \sum_{j=1}^{n_A} E_{i,j}$$

Since each $E_{i,j}$ is the probability of energy transfer event between i^{th} donor and j^{th} acceptor, the sum $\sum_{i=1}^{n_D} \sum_{j=1}^{n_A} E_{i,j} = \sum_{i=1}^{n_D} \sum_{j=1}^{n_A} 1 \cdot E_{i,j}$ is equivalent to the expected number of energy transfer events in the complex given that all donors are excited. The multiplicative factor $1 / n_D$ implies $E_{D,\text{max}}$ measures the expected number of energy transfer events given all donors are excited per donor molecule in the complex.

Informal analysis: A more intuitive understanding can be achieved as follows:

Suppose the complex contains exactly 1 donor and 1 acceptor. In this scenario, the expected number of energy transfer events in the complex is also the expected number of energy transfer events per donor in the complex. Now suppose the FRET efficiency of this pair is 1 (physically unrealistic but let us assume it is possible). This means that every time the donor is excited it will transfer energy to the acceptor - or the expected number of energy transfer events in the complex is 1. Moreover, this also means that the fraction of donor photons lost is 100%. Now suppose the FRET efficiency of the pair is 0 (the chromophores are in such an arrangement). This means the donor will never transfer energy - or the expected number of energy transfer events per donor molecule in the complex is 0. The fraction of donor photons lost is also 0%. So the definitions are equivalent at the extremes and like reasoning could be used to demonstrate that if FRET efficiency is an intermediate value, then the expected number of energy transfer events per lone donor in the complex assuming the donor is excited is equivalent to the fraction of donor photons lost.

Let us now consider what happens if you had 2 donors (D_1 & D_2) and 1 acceptor (A) in the complex. For simplicity, suppose the first donor-acceptor (D_1 - A) pair has a FRET efficiency of 1 and the second donor-acceptor (D_2 - A) pair has an efficiency of 0. This means that every time D_1 is excited it will transfer energy to A though every time D_2 is excited it will fail to transfer energy. So overall, the expected number of energy transfer events in the complex (with both D_1 and D_2) is actually 1 (between D_1 - A). The fraction of donor photons lost though is only 50% since there are two donors (D_1 & D_2) and the only photons lost due to FRET are the ones from D_1 . Thus, the reported FRET efficiency (E_D) is 0.5 which is actually the expected number of energy transfer events per donor in the complex given that both donors were excited. The same reasoning holds true if the individual FRET pairs had intermediate efficiencies as can be deduced from the formal reasoning above.

An abbreviated version of the formal analysis section is provided in Supplementary Note 1.

(2.3) - the authors should explain in greater detail the calculation of MA and MD calibration factors, as these are crucial for the application of the methodology.

Thank you for this suggestion. We have added the following discussion:

“These instrument-specific constants can be computed from *in vitro* measurements of donor (ECFP) and acceptor (EYFP) excitation and emission spectra and specific knowledge of the microscope fluorescence filters as previously described⁹. Briefly, $M_A \approx [\epsilon_A(\lambda)]_{\lambda=430-450 \text{ nm}} \cdot [f_A(\lambda)]_{\lambda=505-575 \text{ nm}} \cdot QY_A$ and $M_D \approx [\epsilon_D(\lambda)]_{\lambda=430-450 \text{ nm}} \cdot [f_D(\lambda)]_{\lambda=505-575 \text{ nm}} \cdot QY_D$. Here, $[\epsilon_A(\lambda)]_{\lambda=430-450 \text{ nm}}$ and $[\epsilon_D(\lambda)]_{\lambda=430-450 \text{ nm}}$ are average values of the molar extinction coefficients of ECFP and EYFP over the excitation bandwidth of FRET filter cube. $[f_D(\lambda)]_{\lambda=505-575 \text{ nm}}$ and $[f_A(\lambda)]_{\lambda=505-575 \text{ nm}}$ are average values of the emission spectra for ECFP and EYFP over the emission bandwidth for the FRET filter cube. Importantly, the both f_A and f_D spectra are normalized such that the total area under each spectrum is 1.”

(2.4) - The analysis of FRET binding isotherms should allow for the recovery of dissociation constants, however these are neither presented or discussed on the basis of literature values. The authors should comment on this.

We omitted discussion of these dissociation constants as we sought to focus on determining interaction stoichiometries from FRET measurements. We have added few sentences at relevant sections.

(Pg. 7) : “In addition, the FRET-binding assays revealed relative dissociation constants ($K_{d,EFF}$) of each IQ truncation to be $800 D_{free}$ units, equivalent to an affinity of $\sim 26 \text{ nM}$ ³⁹ within range of *in vitro* estimates⁴⁰”

(Pg. 9) : “Moreover, these experiments also reveal that apoCaM associates with the holo- Ca_v channel with a relative dissociation constant of $3500 D_{free}$ units ($\sim 115 \text{ nM}$), while Ca^{2+} -bound CaM binds with an enhanced affinity of $820 D_{free}$ units ($\sim 26 \text{ nM}$). While there are no current estimates of CaM binding affinity to holo- Ca_v channels⁴¹, these findings follow trends in *in vitro* affinity measurements of key CaM-binding segments.”

(Pg. 9) : “With regards to relative affinities, like $Ca_v1.2$, $Na_v1.4$ binds to Ca^{2+}/CaM with a substantially higher affinity ($\sim 150 D_{free}$ units) in comparison to Ca^{2+} -free CaM ($\sim 1200 D_{free}$ units).”

For $Na_v1.4$ there are no current estimates of CaM-binding affinity.

This information is summarized as Supplementary Table 1.

Minor Comments:

(2.5) - The format of Reference List should be homogenized.

Thank you. We have modified the reference list accordingly.

(2.6) Pag 3, line 89: Some specific FRET events (as FRET events between identical molecules) might not result in a red shift of fluorescence.

Agreed. Rephrased to: “The excited acceptor may then release a photon, though possibly with a red-shifted spectrum.”

(2.7) Pag 4, line 109: "...quenching of the donor emission spectrum and (2) enhancement in the acceptor emission spectrum" should be replaced by "quenching of the fluorescence intensity of the donor (2) and increase in the fluorescence intensity of the acceptor".

Changed as requested.

(2.8) Page 11, line 302: Rewrite sentence: "In addition, histograms of spatially-resolved apparent donor-centric FRET efficiencies from single cells to deduce the most likely spatial arrangement of fluorophores in the bound complex to infer stoichiometry"

Thank you. Rephrased to – "In addition, histograms of spatially-resolved apparent donor-centric FRET efficiencies from single cells **have been used to** deduce the most likely spatial arrangement of fluorophores in the bound complex to infer stoichiometry."

Supplementary materials:

(2.9) Pag3 and 4: Indexes used for some of the quantities are shown with different values while apparently referring to the same quantity. Namely:

- SYFP(A,500,535) and SYFP(A,500,530LP);
- SFRET(DA,440,535LP) and SFRET(DA,440,535)

Thank you. We have corrected this to be $S_{YFP}(A, 500, 530LP)$ and $S_{FRET}(DA, 440, 535)$ denoting the appropriate FRET cubes used.

(2.10) Pag 6: I_0 is not identified

Thank you. Inserted the following sentence: " I_0 is the average intensity of the excitation source."

Reviewer #3, expert in voltage-gated Ca²⁺ channels (Remarks to the Author):

(3.1) In this study, Ben-Johny et al developed a novel strategy to use FRET signals to determine the stoichiometry of macromolecular complexes. They used the methodology to determine the stoichiometry of calmodulin binding to calcium and sodium channels. Their data show that, at resting Ca, only 1 CaM binds to CaV1.2 channels. However, when Ca increases, 2 CaMs bind to the channel. NaV channels seem to only bind to 1 CaM.

Overall the quality of the data and analyses is excellent. The new method will undoubtedly be useful to many investigators adding to hetero FRET as a means to determine protein stoichiometry.

Thank you for the highly positive assessment of the manuscript.

(3.2) I have a few suggestions. First, the authors should comment on the implications of the study on recent work by Dixon et al (2015) for channel-to-channel interactions, as they propose a 2CaM:1CaV1.2 channel model in which 1 CaM binds in a Ca-dependent manner to the channel.

Thank you for this excellent suggestion. We have now added the following discussion:

"One possibility is that the recruited Ca²⁺/CaM may enable 'functional coupling' of Ca_v1.2⁶⁶. Intriguingly, this study showed that two CaM-dependent Ca_v1.2 functions – the canonical Ca²⁺-dependent inactivation and 'coupled channel gating' – exhibited distinct sensitivities to a CaM inhibitory peptide suggesting that the two functions maybe mediated by two unique CaM⁶⁶.

Given that we detect the binding of two Ca^{2+} /CaM, our findings lend further support to this hypothesis.”

(3.3) It would also be helpful if the stoichiometry of CaM/CaV1.2 complex is determined in a native cell.

While assessing stoichiometry of CaM/Ca_v1.2 in native system would be of some interest, these experiments are highly challenging requiring strategies to ensure 1:1 labeling of fluorophores onto endogenous channels. This is beyond the scope of the present study and will constitute excellent follow up work. Thank you for this great suggestion.

Reviewers' Comments:

Reviewer #2 (Remarks to the Author)

The authors have answered my questions regarding this manuscript. As a final comment regarding the issue of MD and MA coefficients, the authors state in page 19 of the manuscript that the total number of donors and acceptors are defined as CFPest and YFPest, which according to the work of Ericksson and coworkers (2001) is defined as:

$$\text{CFPest} = \text{ND} \cdot \text{Io} \cdot \text{C} \cdot \text{MD}$$

where the total number of donors and acceptors are defined as ND and NA. The authors should correct this and state (as done some paragraphs below already) that these quantities are proportional to the number of donor and acceptor molecules. I also suggest that the relationship between CFPest and ND is included in the manuscript. Failing to do so needlessly obscures the meaning of the data to general readers and just referencing previous works on such a critical point is not enough, especially when the authors offer such a detailed analysis regarding other points. Since this journal is aimed at a general audience, i feel that this clarification is important.

RESPONSE TO REVIEWERS (NCOMMS-16-16204A)

Reviewer #2, (Remarks to the Author):

The authors have answered my questions regarding this manuscript. As a final comment regarding the issue of MD and MA coefficients, the authors state in page 19 of the manuscript that the total number of donors and acceptors are defined as CFP_{EST} and YFP_{EST}, which according to the work of Ericksson and coworkers (2001) is defined as:

$$CFP_{EST} = ND \cdot I_0 \cdot C \cdot MD$$

where the total number of donors and acceptors are defined as ND and NA. The authors should correct this and state (as done some paragraphs below already) that these quantities are proportional to the number of donor and acceptor molecules. I also suggest that the relationship between CFP_{EST} and ND is included in the manuscript. Failing to do so needlessly obscures the meaning of the data to general readers and just referecing previous works on such a critical point is not enough, specially when the authors offer such a detailed analysis regarding other points. Since this journal is aimed at a general audience, i feel that this clarification is important.

We have now included this information regarding CFP_{EST} and YFP_{EST} in the main manuscript as suggested. As outlined in Erickson *et al*, $CFP_{EST} = CFP_{FRET}(440,535,direct)/M_D = N_D \cdot I_0 \cdot C$ and $YFP_{EST} = YFP_{FRET}(440,535,direct)/M_A = N_A \cdot I_0 \cdot C$.

The Methods subsection entitled FRET optical imaging has be modified to:

As detailed in previous publications^{9,69}, CFP_{EST} and YFP_{EST} is proportional to the number of donors (N_D) and acceptors (N_A) given by $CFP_{EST} = N_D \cdot I_0 \cdot C$ and $YFP_{EST} = N_A \cdot I_0 \cdot C$, where I_0 is intensity of the excitation light and C is a proportionality constant (Supplementary Note 1). Experimentally, these values can be determined as,

$$CFP_{EST} = \frac{R_{D1} \cdot S_{CFP}(DA, 440, 480)}{M_D \cdot (1 - E_D)}; \quad YFP_{EST} = \frac{R_{A1} \cdot S_{YFP}(DA, 500, 535)}{M_A} \quad (5)$$